# Autonomous aerial obstacle avoidance using LiDAR sensor fusion

**Qing Liang**[1�], **Zilong Wang**[1�]*, **Yafang Yin**[1], **Wei Xiong**[2], **Jingjing Zhang**[1], **Ziyi Yang**[1]

**1** School Of Electronic Engineering, Xi'an University Of Posts And Telecommunications, Xi'an, Shaanxi Province, China, **2** School Of Information Engineering, Xi'an FANYI University, Xi'an, Shaanxi Province, China

☉ These authors contributed equally to this work.

* wangzilong0203@163.com

**Data Availability Statement:** The experimental LiDAR has a ranging range of 0-10m, a sampling frequency of 10HZ, and each frame of LiDAR data includes 360 data. Due to the large amount of experimental data, we only recorded part of the

## Abstract

The obstacle avoidance problem of unmanned aerial vehicle (UAV) mainly refers to the design of a method that can safely reach the target point from the starting point in an unknown flight environment. In this paper, we mainly propose an obstacle avoidance method composed of three modules: environment perception, algorithm obstacle avoidance and motion control. Our method realizes the function of reasonable and safe obstacle avoidance of UAV in low-altitude complex environments. Firstly, we use the light detection and ranging (LiDAR) sensor to perceive obstacles around the environment. Next, the sensor data is processed by the vector field histogram (VFH) algorithm to output the desired speed of drone flight. Finally, the expected speed value is sent to the quadrotor flight control and realizes autonomous obstacle avoidance flight of the drone. We verify the effectiveness and feasibility of the proposed method in 3D simulation environment.

## 1. Introduction

In recent years, with the development of UAV technology, the low-altitude UAV has been widely used in various fields by virtue of their maneuverability and flexibility. Industrial drones are mainly used in aerial mapping, pesticide spraying and power line patrol. Consumer drones are mainly used for low-altitude aerial photography, cluster performances, and daily entertainment [1]. However, their application environment is mostly low airspace, and the flight environment is complex. Therefore, making the UAV has reliable obstacle avoidance capabilities has become one of the important guarantees for the UAV to safely complete flight tasks in low-altitude complex environments [2].

The obstacle avoidance problem of UAV mainly refers to the design of a method that can make the UAV safely reach the target point from the starting point in the unknown flight environment [3]. In the method, we not only need to consider how to enable the drone to safely avoid obstacles, but also need to meet the requirements of the corresponding flight trajectory and the physical conditions of the drone itself [4]. In the research of UAV obstacle avoidance, according to the classification of UAV on-board sensors, we mainly divide it into five main types, which are based on infrared, ultrasonic, millimeter wave, visual and LiDAR obstacle avoidance methods.

experimental data, and only intercepted 5 frames of LiDAR data. All relevant data are within the paper and its Supporting Information files.

**Funding:** The study was approved by the Key Science and Technology Program of Shaanxi Province (2022GY-094) received by Prof. Yafang Yin. The funders had no role in study design, data collection and analysis, decision to publish, or preparation of the manuscript.

**Competing interests:** We declare that we have no known competing financial interests or personal relationships that could have appeared to influence the work reported in this paper.

In order to realize autonomous intelligent flight of UAV in low altitude complex environment, it is particularly important to research the perception of flight environment and the avoidance of obstacles [5]. Different airborne sensors have different environmental perception degrees and effects [6]. Although the method based on infrared obstacle avoidance is simple in structure, convenient in implementation, and sensitive in response, it has low accuracy, close distance and poor directionality. The ultrasonic obstacle avoidance method has good directivity and is not affected by the color or transparency of objects. However, only using a single ultrasonic sensor can only ensure that the UAV can hover after encountering obstacles, and cannot go around autonomously. The obstacle avoidance method based on millimeter-wave radar can distinguish the size and distance of obstacles with strong directivity and fast response, but the height and accuracy of perceiving obstacles are low, especially the perceived forbidden obstacles. Vision sensors can perceive the color details of multiple obstacles, but lack depth information, and the perception of obstacle distance is weak. The obstacle avoidance method based on LiDAR can obtain complete 3-D information of obstacles, and has strong ability to perceive the distance of obstacles. However, the technology is new and still in the development stage [7].

Based on the research and analysis of different obstacle avoidance methods, in this paper, we utilize an obstacle avoidance method composed of three modules: environment perception module, algorithm obstacle avoidance module, and motion control module [8]. The system framework is shown in Fig 1. Firstly, we use LiDAR sensors to sense obstacles around the flight environment. Next, the sensor data is processed by the VFH algorithm to output the desired speed of drone flight. Finally, the expected speed value is sent to the quadrotor flight control and realize autonomous obstacle avoidance flight of UAV. We verify the effectiveness and reliability of the proposed method in the simulation environment. We summarize our contributions as follows:

1. We utilize an obstacle avoidance method which is composed of environment perception, algorithm avoidance and motion control, and verify its effectiveness and reliability.

2. We not only research the obstacle avoidance algorithm of UAV, but also integrate the algorithm with the hardware, so that the UAV can completely realize autonomous obstacle avoidance flight without human pilot.

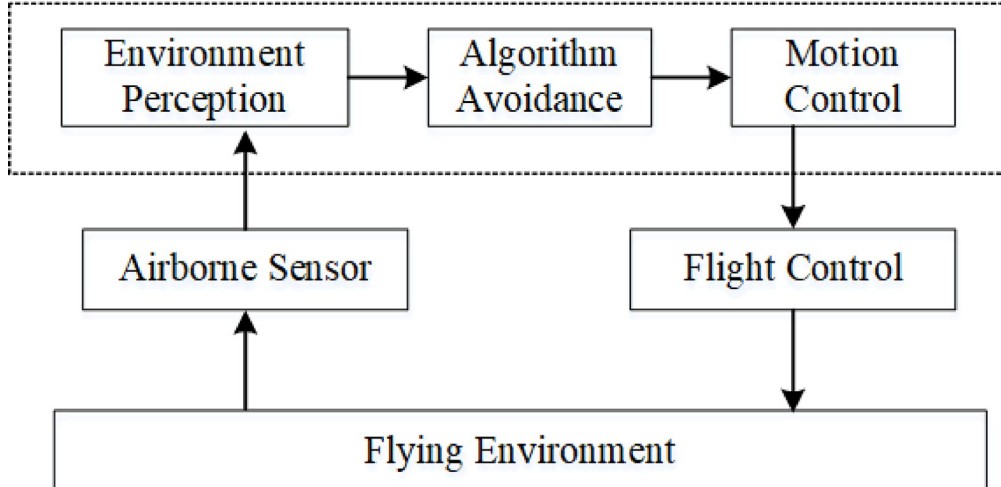

**Fig 1. The framework of UAV obstacle avoidance method.** The framework consists of three parts: environment perception, algorithm avoidance and motion control.

3. Implementation of the proposed method in a combination of software and hardware, and integration with the environment perception mapping module and algorithm obstacle avoidance module in a complete quadrotor platform. Autonomous obstacle avoidance in unknown complex environments is presented.

We discuss relevant literature in section 2, and introduce the system architecture for this work in section 3. Our realization of obstacle avoidance is detailed, and a brief introduction to ranging mechanism of LiDAR and the VFH algorithm in section 4. The implementation details and 3D simulation results about the proposed method, are presented in section 5. Simulation experimental results show fully autonomous obstacle avoidance in unknown complex environments. The paper is concluded in section 6.

## 2. Related works

According to the statistics of UAV cloud data in 2021: UAVs with operating altitudes below 120m accounted for 96.5%. Therefore, the obstacle avoidance problem of low altitude UAV in complex flight environment has attracted the attention and research of scholars at home and abroad. For robotic obstacle avoidance, many methods, ranging from hardware-based sensors to software-based algorithmic applications, have been proposed and applied. Here we provide an overview of representative approaches that are especially relevant to obstacle avoidance for UAV.

To realize autonomous intelligent flight of drones in low altitude complex environment, the perception of the flight environment is particularly important [9]. Different airborne sensors have different environmental perception degrees and effects. Jafri et al. [10] proposes a UAV obstacle avoidance method based on visual and infrared sensor data fusion. In this method, it can quickly perceive the obstacles of the surrounding environment, but it is easily disturbed by light. Jhou et al. [11] applied ultrasonic radar to the problem of robot obstacle avoidance, which solves the problem of not being disturbed by light when perceiving the surrounding environment. However, using only a single ultrasonic sensor can only ensure that the drone hovers after encountering an obstacle, and cannot be go around independently. Another method applying the 3D millimeter wave radar on the issue of drone obstacle avoidance was proposed by Tierney et al. [12], in which it can distinguish the size and distance of obstacles and has strong directivity. However, millimeter-wave radar has low height and accuracy in sensing obstacles, especially forbidden obstacles. Another representative method was developed in [13], where Gao et al. proposed an obstacle detection system based on LiDAR sensor and applied it in unmanned driving system. In this way, the method can obtain complete information of obstacles and has strong perception of obstacles, but the LiDAR technology is new and is still in the development stage.

It is also very important to research the obstacle avoidance algorithm for UAV to realize autonomous obstacle avoidance flight in low altitude complex environment. In the research of obstacle avoidance algorithm, VFH algorithm is widely used in real-time obstacle avoidance of robots. Vector histogram algorithm was first proposed by Borenstein [14] in the field of robot motion. This algorithm is able to detect unknown obstacles and avoid collisions, while steering the mobile robot to a safe target point. However, the algorithm does not take into account the factors of robot motion trajectory. Ulrich and Borenstein [15] improved the vector histogram algorithm, named VFH+, resulting in the robot trajectory more smooth and more reliable. Zhang et al. [16] applied the VFH algorithm to the obstacle avoidance method of RGB-D camera sensor, which greatly improves the smoothness of the robot's motion trajectory and

enhances the obstacle avoidance efficiency. By changing the cost function of VFH, Dong et al. [17] improved the smoothness of the motion trajectory and the efficiency of flight control.

In conclusion, in this paper, we choose LiDAR sensor as environment perception to detect obstacles, and the VFH as obstacle avoidance planning algorithm. At the same time, we utilize a UAV obstacle avoidance method composed of environment perception module, algorithm obstacle avoidance module and motion control module.

## 3. System overview

Low-altitude UAV is small in size and light in weight, making it difficult to carry large-scale sensing equipment to perceive the environment in a wide range. The flight space of UAV is usually in the dense near-earth scene, where multiple security threats coexist, which requires UAV to have higher system response speed and control accuracy. As a result, the design of obstacle avoidance system for low-altitude UAV should make full use of intelligent sensing equipment and operation mode. We can ensure that the low-altitude UAV has the ability of safety guarantee and higher autonomy by choosing sensors with high reliability and obstacle avoidance algorithms with higher computational efficiency [18]. The system architecture is shown in Fig 2. The method is a closed loop system composed of environment perception, obstacle avoidance algorithm and motion control. The environment perception module completes the real-time detection of obstacles; the algorithm obstacle avoidance module completes data processing and outputs the expected flight speed information of the drones; and the motion control module executes the speed output by the algorithm to complete the flying around of the obstacle.

Environment perception module. In the application process of obstacle avoidance of low-altitude UAV, the UAV and the environment need to interact closely and frequently, which requires the UAV to have continuous and accurate perception of the flight environment, that is, the perception of the static flight environment and its own flight positioning. The function of the perception module is to obtain the obstacle information of the flight environment according to the real-time information of the sensor, which is used for collision detection in the flight trajectory. The implementation of this function relies on on-board sensors and processors in low-altitude UAV systems. The module also requires high precision, good consistency and strong real-time performance [19].

Obstacle avoidance algorithm module. In the application of low-altitude UAV in obstacle avoidance task, the UAV needs to interact with the flight path in real time, which requires the UAV to have the ability of avoidance planning. The avoidance planning function of the algorithm usually depends on the perception of the quality of the flight environment, which

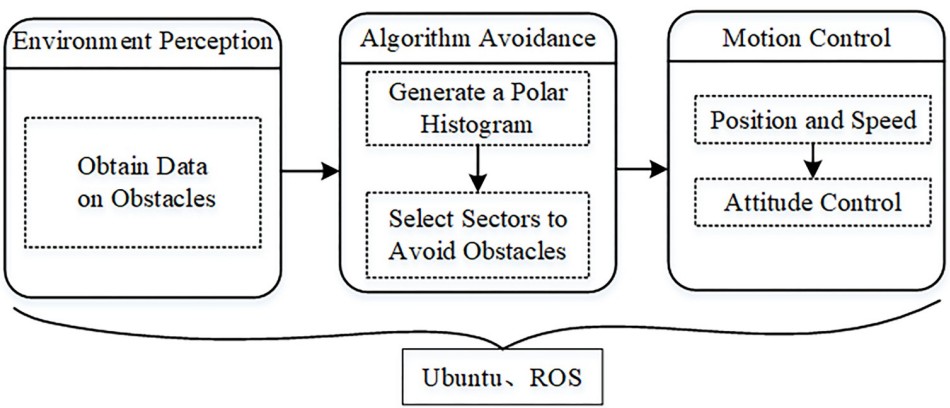

**Fig 2. The complete system diagram.**

determines whether the UAV can reach the target safely. Hence, the module can greatly improve the safe flight capability of the UAV system [20].

Motion control module. Firstly, the airborne sensors perceive the flight environment and the VFH algorithm processes the data in real time [21]. Next, the UAV onboard computer calculates and outputs its position and speed information. Finally, the speed information is sent to the flight control in real time, and then the flight attitude of the UAV is controlled to achieve reasonable and safe obstacle avoidance.

## 4. Realization of obstacle avoidance

### 4.1 Ranging mechanism of LiDAR

As is well-known, the LiDAR is the essential technology to perceive the surrounding environment in all levels of automation, and it can process non-stationary objects in real time. Since LiDAR acts as its own light source, it can also sense its surroundings without interference from light. Our drone is equipped with rotating LiDAR to view the world in 360-degree 3D in real time. This goes far beyond the scope of the cameras, and even beyond the human eye [7]. To illustrate how the LiDAR sensor works, we will highlight a single laser spot. As is shown in Fig 3, a single laser spot is emitted from the sensor, hits an obstacle, and bounces back to the sensor. The reflected data is recorded. We use the time of flight (ToF) to calculate the detection distance.

As is shown in Fig 3, set the delay of laser emission and recovery as $\Delta t$, then the distance between the airborne sensor and the obstacle can be written as:

$$d = \frac{1}{2}C \times \Delta t \tag{1}$$

where C is the speed of light.

Since the exact emission angle $\gamma$ is known, the point coordinate $(x,y)$ of the positioning can be written as:

$$\begin{bmatrix} x \\ y \end{bmatrix} = \begin{bmatrix} d \times \cos\gamma \\ d \times \sin\gamma \end{bmatrix} \tag{2}$$

where $\gamma$ is the emission angle of the LiDAR, $d$ is the measurement distance.

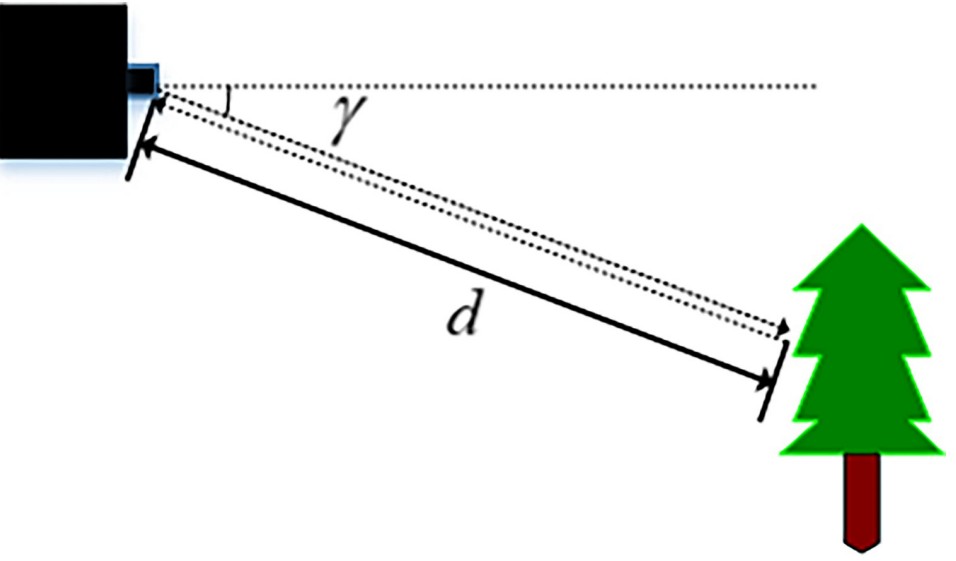

**Fig 3. Principle of single-line LiDAR ranging.**

And then the coordinate values of the scanning points obtained by calculation are saved in the form of polar coordinates.

## 4.2 Principle of VFH algorithm

The vector histogram algorithm measures the distance data between the unknown obstacles and the drone via the airborne sensor, updates the data in real time, and generates a collision-free local path for the drone. The method is mainly completed in two stages. In the first stage, the obstacle data collected by the airborne sensor is simplified into a one-dimensional polar coordinate histogram built around the instantaneous position of the drone. In the second stage, the algorithm selects the sector with low obstacle density from the polar histogram, and then aligns the drone's direction of travel to the direction of this safe sector, that is, to determine the direction of motion [14].

As is shown in Fig 4, the ranging diagram of airborne LiDAR. The obstacle distance information collected by the LiDAR is saved in the form of polar coordinates, and then the VFH algorithm maps the data into a histogram grid graph.

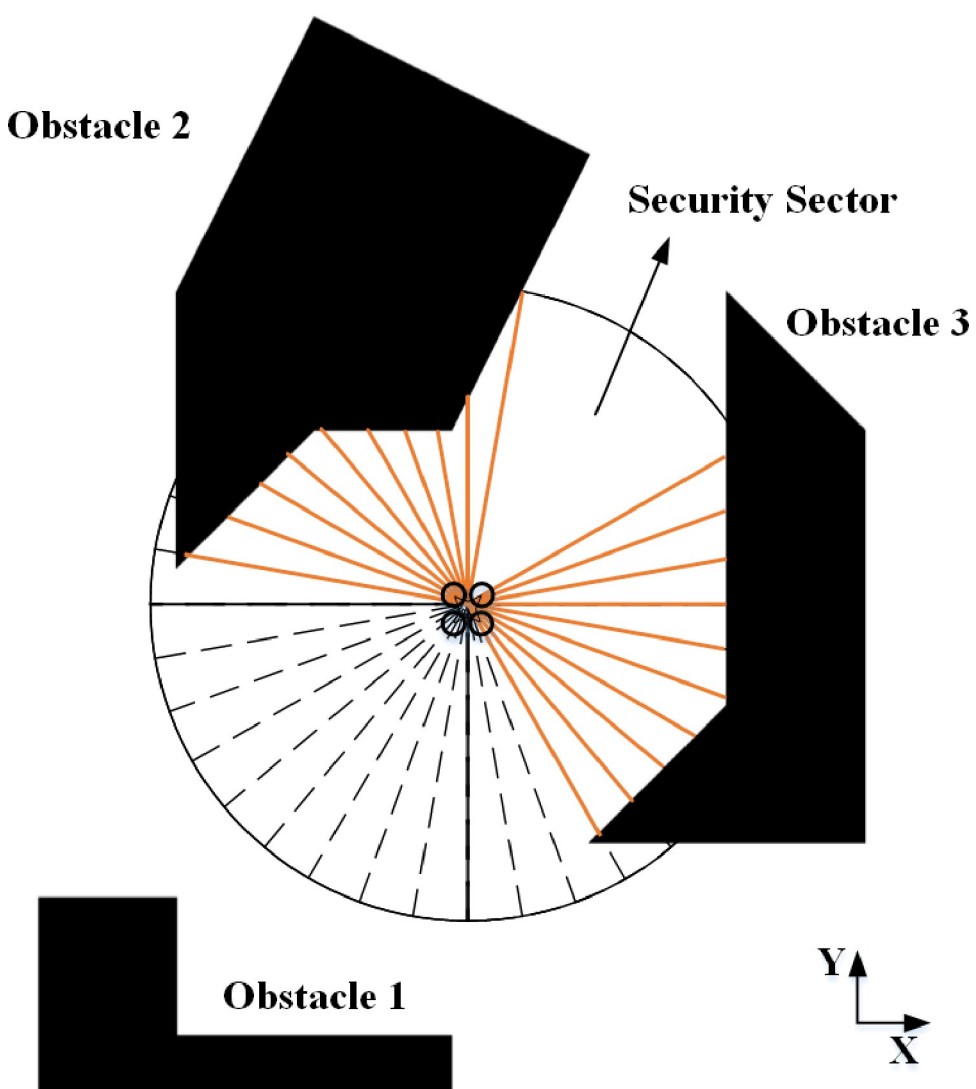

**Fig 4. Airborne LiDAR range sensor.** 36 range sensors equipped to obtain omnidirectional obstacle perception; each one has a sensor view of 10 degrees Black polygons represent obstacles.

A one-dimensional polar coordinate histogram was used to define the motion direction of the drone. The active cell was defined the drone center point (DCP). The active cell moved with it, and each moving step covered a new area.

The contents of each active cell $(i, j)$ in the histogram grid will be treated as an obstacle vector whose direction $\beta$ is determined by DCP. Then the formulation for $\beta_{i,j}$ is written as

$$\beta_{i,j} = \tan^{-1}(\frac{y_i - y_0}{x_i - x_0}) \tag{3}$$

and the magnitude $m_{i,j}$ is given by

$$m_{i,j} = (c_{i,j}^*)^2 (a - bd_{i,j}) \tag{4}$$

where $a,b$ are positive constants, $\beta_{i,j}$ is the direction from active cell $(i, j)$ to the DCP, $(x_0, y_0)$ is the present coordinates of the DCP, $(x_i, y_i)$ is the coordinates of active cell $(i, j)$, $c_{i,j}^*$ is the certainty value of active cell $(i, j)$, the square here expresses to reduce noise which caused by single occurrence of senor detection. And $d_{i,j}$ is the distance between active cell $(i, j)$ and the DCP.

$k$ is the sector corresponding to the moving unit of the obstacle, and the discrete angle corresponding to each sector $k$ is quantized as the multiple of the angular resolution $\alpha$ of the airborne sensor, which is expressed as:

$$k = INT(\beta_{i,j}/\alpha) \tag{5}$$

$h_k$ is the polar obstacle density which represents the probability of encountering an obstacle in the direction of the sector as:

$$h_k = \sum_{i,j} m_{i,j} \tag{6}$$

Since the distribution of polar barrier density is discrete, which may lead to uneven obstacle estimation. Therefore, the smooth polar obstacle density $h_k'$ is obtained by applying the function as:

$$h_k' = \frac{h_{k-1} + 2h_{k-l+1} + \cdots + lh_k + \cdots + 2h_{k+l-1} + h_{k+1}}{2l + 1} \tag{7}$$

where $n$ is the number of sectors. $l$ is a constant integer, chosen by experiment or simulation.

After data processing by algorithm, the polar histogram of environmental obstacles can be obtained, as is shown in Fig 5.

The entire sector of the polar histogram is divided into unsafe and safe valleys by the threshold value $\eta_1$. In order to determine the optimal direction of the drone, the safety valley is divided into narrow valley and wide valley by threshold value $\eta_2$.

In the VFH algorithm, the cost function we adopt is shown as follows:

$$g(c) = \mu_1 \cdot \triangle(c, k_t) + \mu_2 \cdot \triangle(c, \frac{\theta_i}{\alpha}) + \mu_3 \cdot \triangle(c, k_{n,i-1}) \tag{8}$$

where, $\triangle(c_1, c_2)$ is a function that computes the absolute angle difference between two sectors $c_1$ and $c_2$. $\triangle(c, k_t)$ refers to the target direction, $\triangle(c, \frac{\theta_i}{\alpha})$ refers to the current direction of motion and $\triangle(c, k_{n,i-1})$ is the previously selected direction of motion.

In the cost function, the first term represents the difference between the candidate direction and the target direction. The larger the difference is, the more times the candidate direction guides the drone away from the target direction. The second term represents the difference between the candidate direction and the robot's current direction. The bigger the difference,

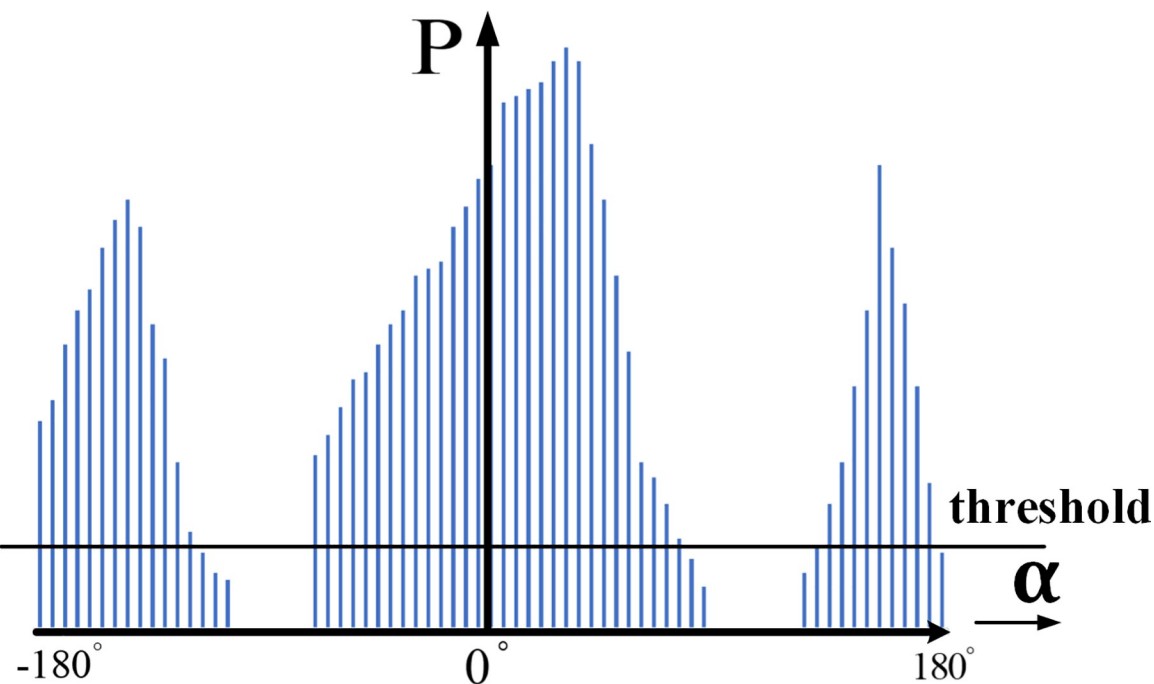

**Fig 5. Polar histogram.** The X axis is the angle of the obstacle perceived by the drone. The Y axis represents the probability of an obstacle in that direction.

the more direction of motion we have to change. The third term represents the difference from the candidate direction and the previously selected direction of motion. The larger the value, the greater the change in the new turn command.

The higher $\mu_1$ is, the more goal-oriented the robot's behavior. The higher $\mu_2$ is, the more the robot tries to execute an efficient path with a minimum change of direction of motion. The higher $\mu_3$ is, the more the robot tries to head towards the previously selected direction and the smoother is the trajectory.

The second and third terms of the cost function are used to determine the smoothness of the robot's motion trajectory. We hope that the drone obstacle avoidance is goal-oriented, so the weight setting of the cost function needs to meet the requirement of $\mu_1 > \mu_2 + \mu_3$.

In practice, the polar histogram will be used to firstly identify all the gaps that are large enough to allow the drone to pass through. Next the cost function is calculated for all these gaps, and finally the path with the lowest cost function is selected to pass through. The cost function is affected by three factors: the target direction, the current direction of the drone, and the previously selected direction. The resulting cost is the weighted value of these three factors. We can adjust the preference of the drone by adjusting different weights.

### 4.3 Matlab simulation

Computer simulation is carried out for the VFH algorithm using in Matlab. The algorithm simulation flow chart of VFH is shown in Fig 6.

In this two-dimensional simulation, it is assumed that the drone is equipped with a scanning laser range sensor with a field of view of 360˚. Also, its location is known and only the kinematic motion of the drone is considered. The laser range sensor is simulated in Matlab, and 36 range data are returned in each sampling period. The rate about simulation loop is 10 Hz. As is shown in Fig 7.

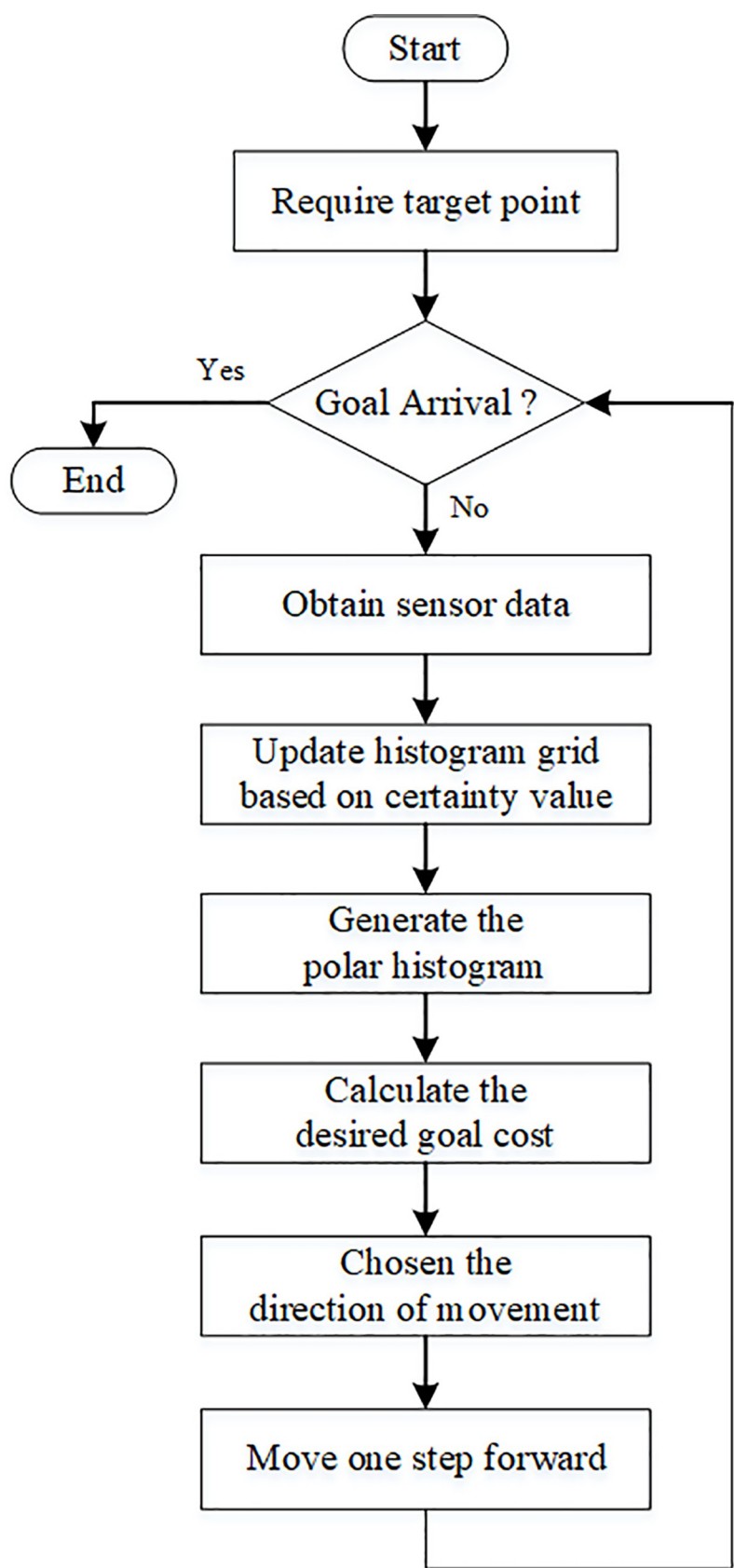

**Fig 6. VFH algorithm flow chart.** The rate about simulation loop is 10 Hz.

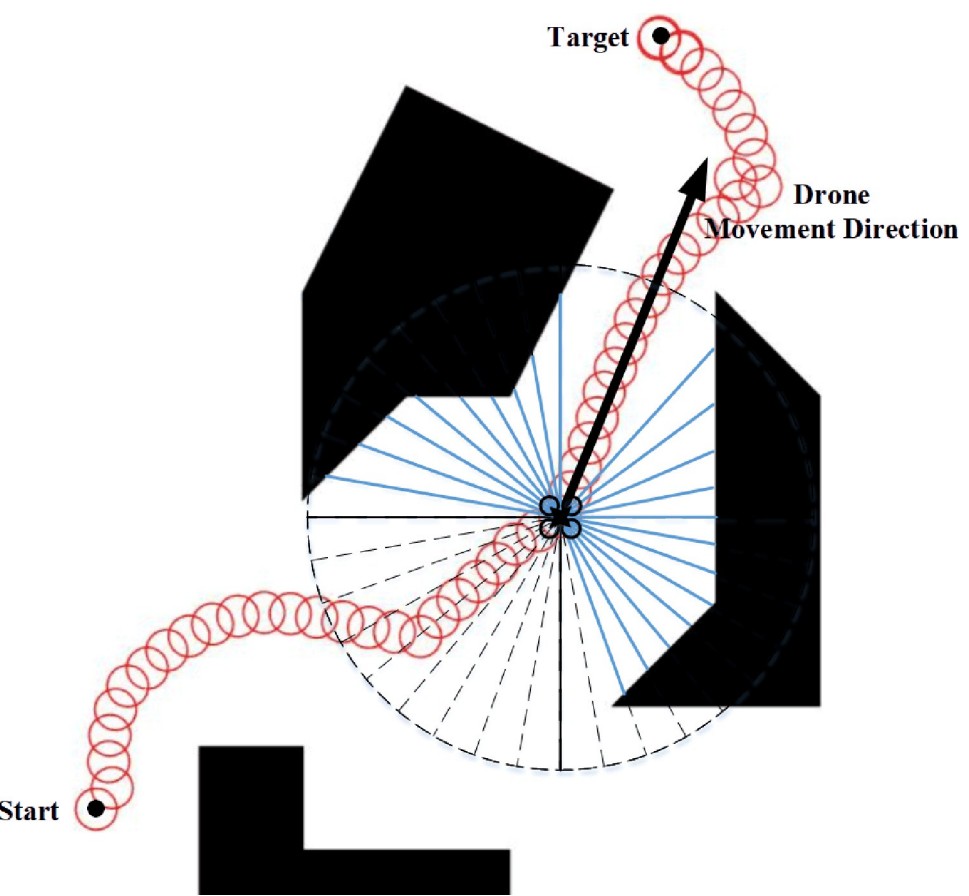

**Fig 7. Simulation method.** The FOV of LiDAR sensors is divided into 36 sectors. The solid blue line denoted obstacle distances. The red circle represents the trajectory of drones. It is assumed that the range finder always aligned with the drone's direction of motion in each simulation loop.

In this simulation, the size of the drone is a circle with a radius of 10 cm, and the moving speed is 1 m/s. As is shown in Fig 8, the drone successfully passes through the obstacles of the unknown environment with a relatively smooth path. The drone obstacle avoidance system mainly consists of three parts: obstacle perception, algorithm avoidance and obstacle flying around. First of all, in the actual obstacle avoidance application, we set a desired flight path for the drone and specify the flight target. Then, the airborne sensor data is transmitted to the airborne computer for algorithm processing, and the expected position and speed information of is output. Finally, the speed information is sent to the flight control system in real time, so as to realize autonomous obstacle avoidance flight.

## 5. Implementation details and results

### 5.1 System settings

Our obstacle avoidance algorithm module is implemented in C++, and the flight simulation experiments are based on Ubuntu 18.04 system. The computing resource include the computer which has a multi-core Intel Core i5-12500H processor running at 4.50 GHz with 8 GB RAM. Autonomous obstacle avoidance flight experiments are performed in an unknown environment using a single LiDAR sensor. The field of view (FOV) of LiDAR is 360 degrees and the scanning frequency is 10Hz. The flight control system of the drone adopts a virtual PX4 controller.

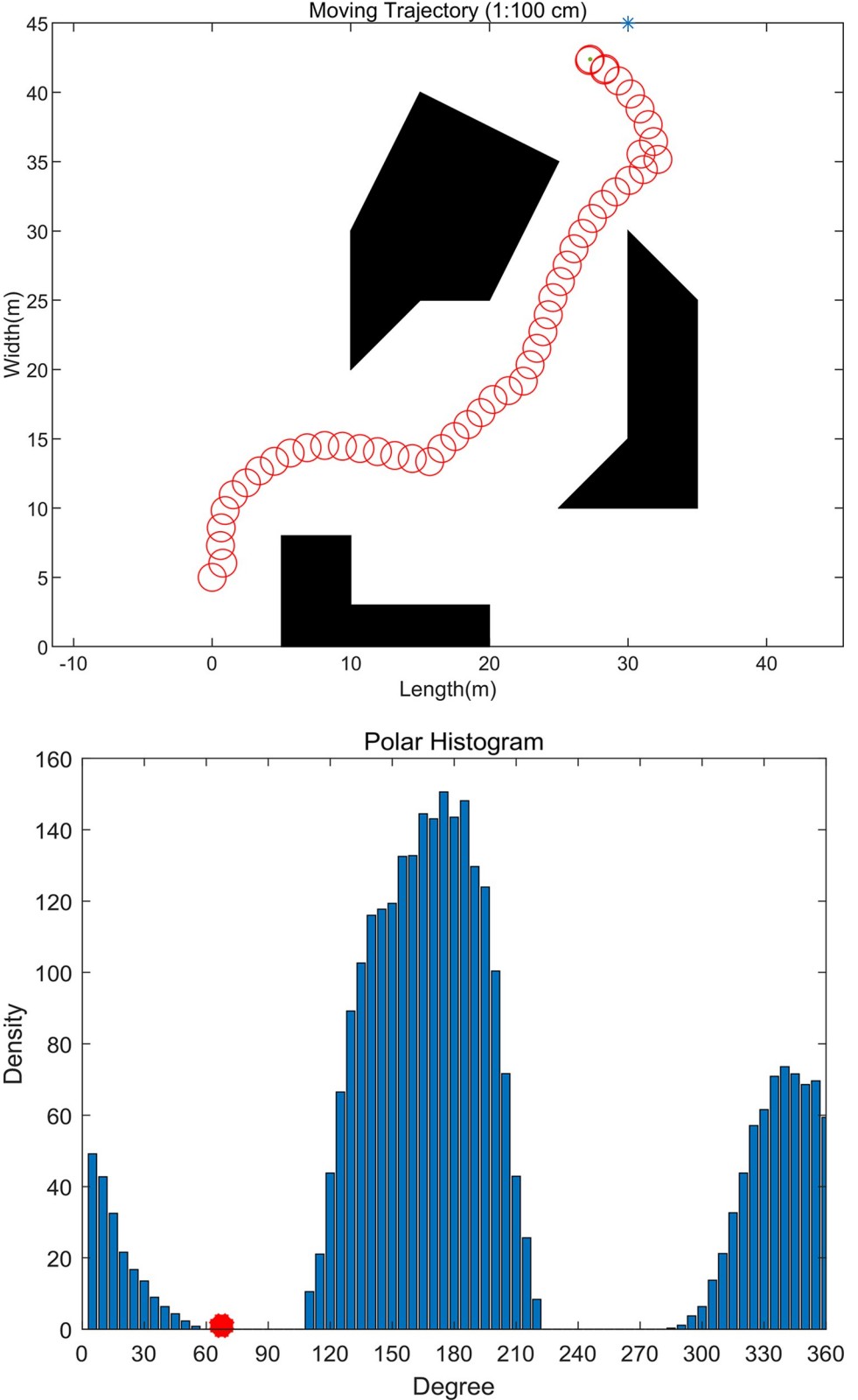

**Fig 8. Simulation result of VFH algorithm.** (a) Obstacle avoidance path of UAV movement; (b) Polar histogram of obstacles at a given time.

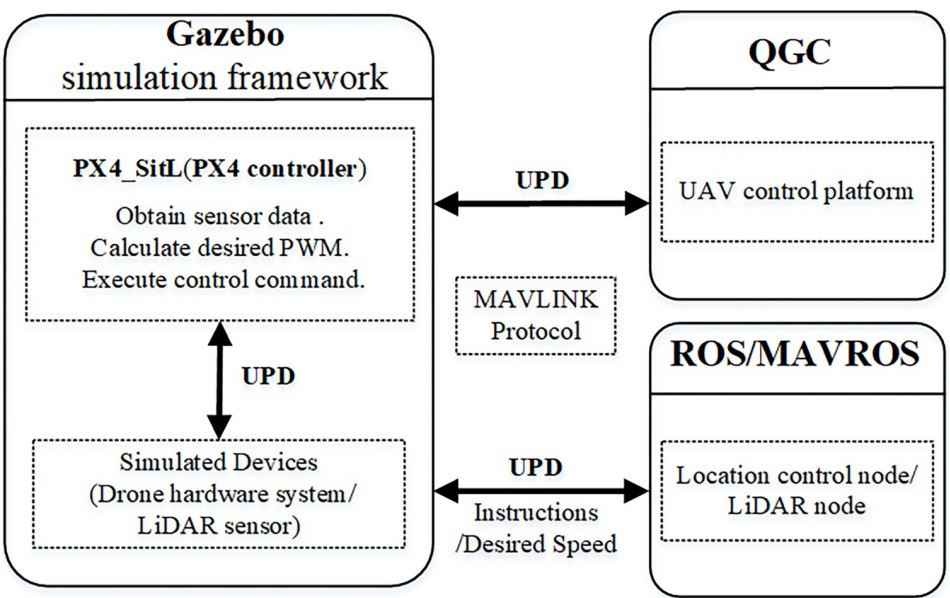

**Fig 9. ROS simulation architecture.** User Datagram Protocol (UPD) is a data transfer protocol.

## 5.2 Simulation system architecture

The simulation environment software architecture mainly consists of three parts: QGound-Controller (QGC), Gazebo virtual environment simulation and ROS system data transmission. The simulation software architecture is shown in Fig 9. ROS, as a popular robot development platform, cooperates with Gazebo simulation environment [22]. PX4 specifically provides an analog controller PX4-StiL that runs in a ROS environment. Firstly, QGC data and ROS node data information are transmitted to the Gazebo simulation platform via the MAVLINK protocol. Next, the Gazebo simulation platform processes the acquired data, calculates the desired control signal and executes it. Finally, drone obstacle avoidance visualization was realized on Gazebo. Finally, the visualization of UAV obstacle avoidance is realized on Gazebo.

## 5.3 Construction of flight simulation environment

Gazebo provides the ability to accurately and efficiently simulate robot motion in complex indoor and outdoor environments. It allows robots to be modeled, simulate their physical behavior, render them in 3D and publish their sensor data. What's more, it's easy to integrate in the ROS ecosystem. Gazebo can use multiple components to simulate an environment. Firstly, a hierarchy of robot components (joints, links, and sensors) is defined through a model file on Gazebo. Secondly, it provides real-world files to place real-world elements such as cars, buildings, and trees. Sensors also respond to these elements, such as LiDAR, which detects the distance of obstacles and records elements of the surrounding environment [23]. In Gazebo, we built a 80m×80m×10m simulated 3D environment, a forest, in which obstacles were randomly placed, as is shown in Fig 10.

## 5.4 Results

We put the drone model in the 3D forest simulation environment. At the QGC ground control station, we set the starting and target points for the drone, and start the software algorithm to let the drone pass through the forest environment. At the QGC ground control station, we can

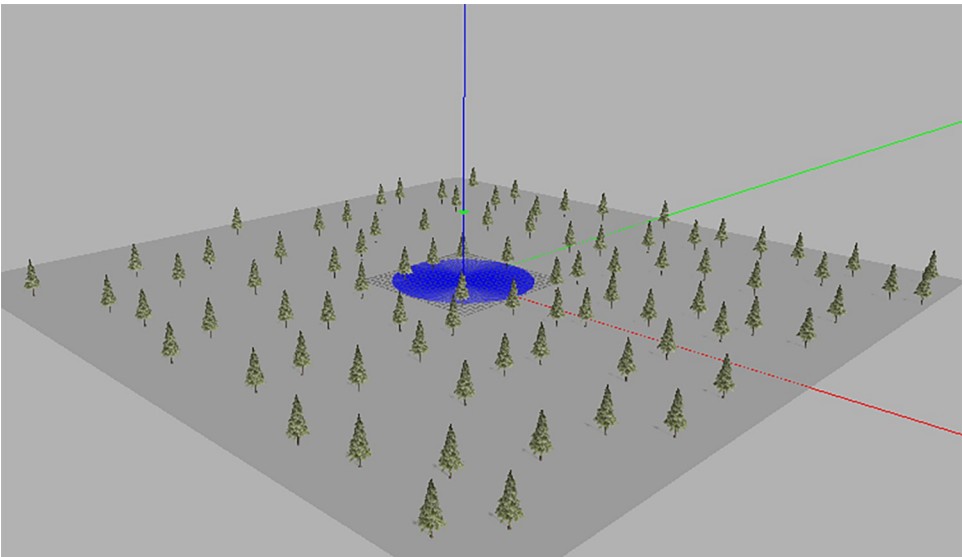

**Fig 10. 3D simulation of the flight environment.**

get the real-time flight trajectory of the drone. The brown straight line is our artificially planned initial drone trajectory route, and the red line is the real-time flight trajectory of the drone as it moves through the forest, as is shown in Fig 11. From the experimental results, it can be seen that the drone can accurately identify obstacles and fly around them, and finally reach the set target point safely.

In the flying environment with obstacles, we set 5 waypoints and set the flight altitude of the drone to 2m. In the three cases of drone obstacle avoidance flight speed of 2m/s, 1m/s and 0.5m/s respectively, we conducted an experiment each. The real-time obstacle avoidance flight trajectory is shown in Fig 11.

From the experimental results, it can be seen that the local flight trajectory is effectively smoothed as the expected speed of the drone's obstacle avoidance flight decreases. By adjusting the weight parameters of the cost function, we can change the choice preference of the drone's movement direction, adjust the expected speed of the obstacle-avoidance flight and optimize

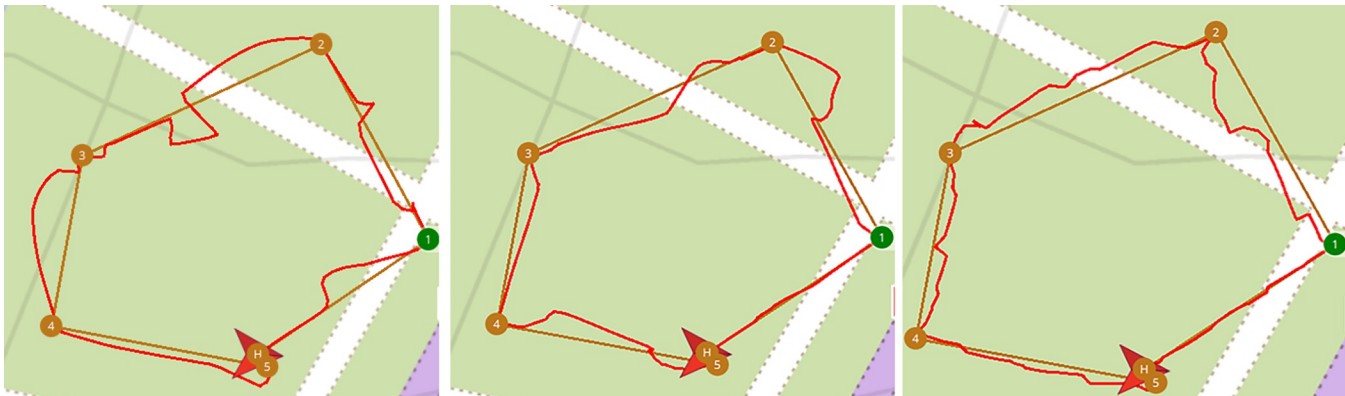

**Fig 11. UAV obstacle avoidance trajectory.** The brown straight line is the initial drone route for our artificial planning. The red line represents the real-time flight trajectory of the drone through the forest. (a) The trajectory with a speed of 2m/s; (b) The trajectory with a speed of 1 m/s; (c) The trajectory with a speed of 0.5 m/s.

the flight path. Since the VFH algorithm is a local optimization algorithm, the local flight path can be smoothed, but the smoothing effect of the global flight path is not as expected. We plan to add global path planning algorithms, such as the A* algorithm, to the obstacle avoidance system to optimize the global flight trajectory of the drone.

## 6. Conclusion and future work

In this paper, we mainly propose the obstacle avoidance method of fusion of algorithm and hardware sensor device, which integrates the results of obstacle perception by on-board sensors with obstacle avoidance algorithms. The proposed method realizes reasonable and safe obstacle avoidance of quadrotor in low altitude complex environment. Firstly, we use LiDAR sensors to percept obstacles around the environment. Then, the environmental data collected by the LiDAR sensor is processed by the VFH algorithm to output the desired speed of the drone flight. Finally, the expected speed value is sent to the drone flight controller to realize autonomous obstacle avoidance flight of drone. We present fully autonomous obstacle avoidance flights in 3D simulation unknown environment to validate our method.

For quadrotor obstacle avoidance, since we did not consider the optimal flight path of the drone, the flight trajectory was not smooth, and excessive on-board computer computing resources are wasted. To solve this problem, in the future, we intend to add a path planning module and navigation technology to make the flight path of UAV smoother and improve its obstacle avoidance efficiency.

## Supporting information

**S1 Dataset.**
(XLSX)

## Author Contributions

**Conceptualization:** Qing Liang.

**Data curation:** Zilong Wang.

**Formal analysis:** Zilong Wang, Yafang Yin, Ziyi Yang.

**Investigation:** Zilong Wang.

**Methodology:** Yafang Yin, Jingjing Zhang.

**Software:** Zilong Wang.

**Supervision:** Yafang Yin, Wei Xiong.

**Writing – original draft:** Zilong Wang.

**Writing – review & editing:** Zilong Wang.

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
