## [Decision Letter · Decision Letter 0]

18 Apr 2023

PONE-D-23-08540Autonomous Aerial Obstacle Avoidance Using LiDAR Sensor FusionPLOS ONE

Dear Dr. Wang,

Thank you for submitting your manuscript to PLOS ONE. After careful consideration, we feel that it has merit but does not fully meet PLOS ONE’s publication criteria as it currently stands. Therefore, we invite you to submit a revised version of the manuscript that addresses the points raised during the review process.

We look forward to receiving your revised manuscript.

Kind regards,

Farzan Majeed Noori

Academic Editor

PLOS ONE

Journal Requirements:

   "No"

7. We note that Figures 8 and 10 in your submission contain [map/satellite] images which may be copyrighted. All PLOS content is published under the Creative Commons Attribution License (CC BY 4.0), which means that the manuscript, images, and Supporting Information files will be freely available online, and any third party is permitted to access, download, copy, distribute, and use these materials in any way, even commercially, with proper attribution. For these reasons, we cannot publish previously copyrighted maps or satellite images created using proprietary data, such as Google software (Google Maps, Street View, and Earth). For more information, see our copyright guidelines: http://journals.plos.org/plosone/s/licenses-and-copyright.

a. You may seek permission from the original copyright holder of Figures 8 and 10 to publish the content specifically under the CC BY 4.0 license.  

8. Please remove your figures (Figure 8) from within your manuscript file, leaving only the individual TIFF/EPS image files, uploaded separately. These will be automatically included in the reviewers’ PDF.

Reviewers' comments:

Reviewer's Responses to Questions

**Comments to the Author**

1. Is the manuscript technically sound, and do the data support the conclusions?

Reviewer #1: Partly

Reviewer #2: Yes

2. Has the statistical analysis been performed appropriately and rigorously? 

Reviewer #1: N/A

Reviewer #2: No

3. Have the authors made all data underlying the findings in their manuscript fully available?

Reviewer #1: Yes

Reviewer #2: Yes

4. Is the manuscript presented in an intelligible fashion and written in standard English?

Reviewer #1: Yes

Reviewer #2: Yes

5. Review Comments to the Author

Reviewer #1: In the motion block of Fig. 2, a spelling correction for "speed" is required. Please make sure to correct the spelling error to maintain accuracy and professionalism in the paper.

It would be helpful if the authors could show the calculation of the path cost function as discussed in Section 4.2. Providing a clear and detailed explanation of the method used to calculate the path cost function would enhance the reproducibility of the results.

Figure 10, which is referred to in the Results section, is missing. Please make sure to include the missing figure to ensure that the readers have access to all the necessary information for a complete understanding of the results and findings.

The optimization of flight path for the drone and the smooth flight trajectory are crucial factors in supporting the proposed method. It would be beneficial if the authors could provide alternative trajectories for simulation purposes to further demonstrate the impact of the proposed method.

Overall, the paper presents interesting research on the optimization of flight path for drones. However, attention to details such as spelling errors, missing figures, and providing clear explanations of the methodology would enhance the quality and comprehensibility of the paper. Additionally, providing alternative trajectories for simulation purposes would further strengthen the support for the proposed method.

Reviewer #2: the word document doesn't have figures in there places, only Fig 8 was in its place. Rest were not.

Add complete flow diagram of the algorithm that is used in " 4.1 Realization of Obstacle Avoidance", as I am not able to see it for better understanding of operation used to perform this experimentation.

6. PLOS authors have the option to publish the peer review history of their article (what does this mean?). If published, this will include your full peer review and any attached files.

Reviewer #1: No

Reviewer #2: No

While revising your submission, please upload your figure files to the Preflight Analysis and Conversion Engine (PACE) digital diagnostic tool, https://pacev2.apexcovantage.com/. PACE helps ensure that figures meet PLOS requirements. To use PACE, you must first register as a user. Registration is free. Then, login and navigate to the UPLOAD tab, where you will find detailed instructions on how to use the tool. If you encounter any issues or have any questions when using PACE, please email PLOS at figures@plos.org. Please note that Supporting Information files do not need this step.<quillbot-extension-portal></quillbot-extension-portal>

---

## [Author Response · Author response to Decision Letter 0]

16 May 2023

Reviewer #1:

First of all, thank you very much for your review of our paper and your comments, and then I will respond to your questions individually.

(1) The spelling error in Fig 1 has been corrected.

(2) In section 4.2 of the paper, the cost function calculation formula and detailed description of the local path planning of the drone have been added.

(3) In section 5.4 of the paper, experimental data with a flight speed of 0.5m/s of the drone have been completed. At the same time, the experimental data of the drone flying speed of 2m/s and 1m/s are added, and the three experimental results are compared and explained.

(4) Since the VFH algorithm is a local obstacle avoidance algorithm, we can optimize the local flight trajectory of the drone by adjusting the cost function of the algorithm, but the effect of global flight trajectory optimization is not as expected. In section 5.4 of the paper, we add corresponding experimental data to explain this problem. In the future, we plan to introduce a global path planning algorithm module to this obstacle avoidance system, such as the A* algorithm, to optimize the global flight trajectory of the drone.

Reviewer #2:

First of all, thank you very much for your review of our paper and your comments, and then I will respond to your questions individually.

(1) The ROS simulation architecture of section 5.4 has been modified to remove the non-conforming image.

(2) In section 4.3 of the paper, Fig 6 - flowchart of VFH algorithm simulation has been added.

---

## [Editor Report · Decision Letter 1]

1 Jun 2023

Autonomous Aerial Obstacle Avoidance Using LiDAR Sensor Fusion

PONE-D-23-08540R1

Dear Dr. Wang,

Thanks for revising the manuscript and updating it based on reviewers comments.

We’re pleased to inform you that your manuscript has been judged scientifically suitable for publication and will be formally accepted for publication once it meets all outstanding technical requirements.

Kind regards,

Farzan Majeed Noori

Academic Editor

PLOS ONE

Additional Editor Comments (optional):

Reviewers' comments:

<quillbot-extension-portal></quillbot-extension-portal>

---

## [Editor Report · Acceptance letter]

19 Jun 2023

PONE-D-23-08540R1 

Autonomous Aerial Obstacle Avoidance Using LiDAR Sensor Fusion 

Dear Dr. Wang:

I'm pleased to inform you that your manuscript has been deemed suitable for publication in PLOS ONE. Congratulations! Your manuscript is now with our production department. 

Kind regards, 

on behalf of

Dr Farzan Majeed Noori 

Academic Editor

PLOS ONE